# Comparing the Influence of Assembly Processes Governing Bacterial Community Succession Based on DNA and RNA Data

**DOI:** 10.3390/microorganisms8060798

**Published:** 2020-05-26

**Authors:** Xiu Jia, Francisco Dini-Andreote, Joana Falcão Salles

**Affiliations:** 1Microbial Ecology Cluster, Genomics Research in Ecology and Evolution in Nature (GREEN), Groningen Institute for Evolutionary Life Sciences (GELIFES), University of Groningen, 9747 AG Groningen, The Netherlands; 2Department of Plant Science, The Pennsylvania State University, University Park, PA 16802, USA; andreote@psu.edu; 3Huck Institute of Life Sciences, The Pennsylvania State University, University Park, PA 16802, USA

**Keywords:** 16S rRNA gene, community turnover, ecological modeling, selection, dispersal, drift

## Abstract

Quantifying which assembly processes structure microbiomes can assist prediction, manipulation, and engineering of community outcomes. However, the relative importance of these processes might depend on whether DNA or RNA are used, as they differ in stability. We hypothesized that RNA-inferred community responses to (a)biotic fluctuations are faster than those inferred by DNA; the relative influence of variable selection is stronger in RNA-inferred communities (environmental factors are spatiotemporally heterogeneous), whereas homogeneous selection largely influences DNA-inferred communities (environmental filters are constant). To test these hypotheses, we characterized soil bacterial communities by sequencing both 16S rRNA amplicons from the extracted DNA and RNA transcripts across distinct stages of soil primary succession and quantified the relative influence of each assembly process using ecological null model analysis. Our results revealed that variations in α-diversity and temporal turnover were higher in RNA- than in DNA-inferred communities across successional stages, albeit there was a similar community composition; in line with our hypotheses, the assembly of RNA-inferred community was more closely associated with environmental variability (variable selection) than using the standard DNA-based approach, which was largely influenced by homogeneous selection. This study illustrates the need for benchmarking approaches to properly elucidate how community assembly processes structure microbial communities.

## 1. Introduction 

Research on ecological succession is key to advance our understanding of how communities are assembled and affect ecosystem functioning and dynamics. In accordance with classical studies on plant community succession, the relatively recent advances in high-throughput sequencing technologies and microbiome profiling have revealed that microbial communities across a broad range of habitats also undergo sequential changes through different time scales [1,2,3].

Most studies have profiled bacterial community composition across succession gradients by sequencing 16S rRNA gene from environmental DNA [3,4,5,6]. However, the environmental DNA used for identifying bacterial taxonomy/composition can not only come from viable cells, but also from extracellular DNA and undecomposed DNA from dead cells (termed as ‘relic DNA’), which is known to be ubiquitous in soils [7,8]. Due to the fact that relic DNA can stay in the environment for an unpredicted period of time, profiling bacterial communities by DNA not only has the risk of over inflating the diversity estimation [9], but also can lead to potentially erroneous signals of temporal variability in microbial communities [10]. It was argued that relic DNA contribute minimally to the characterization of bacterial community composition, but bias can still arise when community turnover is faster than the turnover of the relic DNA pool [11]. Since the sequential change of the bacterial community over time is the interest of studies on ecological succession, relic DNA left in the environmental DNA samples will obscure these changes. Unlike DNA, RNA has short half-life times, and can often be used to represent putatively active fractions of bacterial taxa that are alive in the environment [12]. As such, the RNA-based approach can better reflect the turnover of bacterial communities during succession, especially for short term successional dynamics. However, few studies have examined bacterial community succession using both DNA- and RNA-based approaches [13,14,15], and we still lack understanding of how much those two approaches affect the outcome of bacterial community turnover during succession.

Disentangling the relative influences of community assembly processes structuring the microbiome distribution is important for understanding community turnover during succession [16]. For instance, the trajectories of bacterial communities during primary and secondary succession have been shown to follow different assembly processes [16]. In brief, during primary succession, the assembly processes were found to gradually shift from stochastic to deterministic [16]. Conversely, during secondary succession, the assembly processes governing the dynamics of bacterial communities were found to be influenced by the previous state of the communities prior to a disturbance event [16]. For instance, soil bacterial communities were structured from more to less stochastic processes after a wildfire disturbance [17]. While a disturbance experiment has shown a strong deterministic recovery during secondary succession [15].

Quantifying the assembly processes is often dependent on either taxonomic or phylogenetic information of the investigated meta-community. Interestingly, it was shown that the phylogenetic structure of RNA-inferred bacterial communities is significantly more clustered than that of the DNA-inferred communities, thus resulting in a stronger environmental filtering signal [18]. Selection through environmental filter or biotic interactions allows species with certain traits to establish and persist within the local community. The selective pressure can be either evenly distributed among communities (i.e., homogeneous selection) or heterogeneous (i.e., variable selection), resulting in community turnover and variation differences [16,19]. Given that DNA and RNA differ in stability, we expect RNA-inferred communities response to biotic/abiotic fluctuations to be faster than those inferred by DNA. Therefore, the signal of variable selection is expected to be stronger in RNA-inferred communities, while the signal of homogeneous selection is expected to be stronger in DNA-inferred communities.

In this study, we investigated the dynamics of bacterial communities across five successional stages in a primary succession soil chronosequence. Soil samples were collected across the successional stages and within successional stages at four time points. The analysis of DNA-derived 16S rRNA gene sequences encompassed the total fraction of bacterial communities including DNA from non-viable cells; and the RNA-derived 16S rRNA sequences encompassed the content of bacterial ribosomes indicating the potentially active fraction of total bacteria communities. We first examined to what extent these two approaches affect the results of bacterial community turnover during primary succession, and then compared the interplay of community assembly processes, and evaluated whether these two approaches affect the outcomes and conclusions of microbial community successional dynamics in soils.

## 2. Materials and Methods 

### 2.1. Study Site and Soil Sampling

The study was conducted in a salt marsh ecosystem at the island of Schiermonnikoog, the Netherlands (53°30′ N, 6°10′ E). This island displays a progressive growth expansion eastwards caused by the continuous sedimentation of particles carried by wind, currents and flooding cycles. This natural chronosequence spans over 100 years of succession, in which early successional stages are located in the east and late stages in the west. Soil physicochemical properties, aboveground biomass, and ecosystem disturbances gradually change from early to late successional stages [20,21]. For instance, the flooding frequency, soil sand content and soil pH decrease as succession proceeds, while vegetation coverage, soil silt and clay content and overall nutrient status (e.g., total nitrogen, total carbon, nitrate and ammonia) increase [21]. A detailed description of the sampling sites can be found in Dini-Andreote et al. [21].

To characterize the spatiotemporal variation of bacterial communities in this system, triplicate soil samples were collected from five successional stages (i.e., 0, 10, 40, 70 and 110 years) in May, July, September and November 2017 (60 samples in total). At each replicate, 20 soil cores (3.5 cm diameter, 10 cm depth) were randomly sampled. A total of 2 g of the collected soil that was homogenously mixed was preserved in LifeGuard Soil Preservation Solution (Qiagen, Hilden, Germany) and stored at −80 °C for further nucleic acid extraction.

### 2.2. Nucleic Acid Extraction, Amplicon Library Preparation and Sequencing

The total RNA and DNA were co-extracted from 2 g of soil for each of the 60 samples using the RNeasy PowerSoil Total RNA kit (Qiagen, Hilden, Germany) with the RNeasy PowerSoil DNA Elution kit (Qiagen, Hilden, Germany), following the manufacturer’s instructions (Figure 1). After eluting the RNA samples from the capture column of the RNeasy PowerSoil Total RNA kit, the bound DNA on the capture column was further eluted using the RNeasy PowerSoil DNA Elution kit. Remaining DNA in RNA samples was removed using the DNase Max kit (Qiagen, Hilden, Germany). We performed PCR reactions for 10% of RNA samples to verify whether the amount of DNase applied was enough to ensure DNA was completely removed from RNA samples. The DNA-free RNA was converted to cDNA by incubating with random hexamers using the Transcriptor High Fidelity cDNA Synthesis Kit (Roche, Basel, Switzerland), which was then purified using the MinElute PCR Purification Kit (Qiagen, Hilden, Germany). DNA and cDNA were quantified using a NanoDrop 2000 Spectrophotometer (Thermo Fisher scientific, Waltham, MA, USA).

We profiled the DNA- and RNA-inferred bacterial communities by sequencing the V4 region of the bacterial 16S rRNA gene and 16S rRNA transcripts, respectively, using primer pair 515F (5′-GTGCCAGCMGCCGCGGTAA-3′) and 806R (5′-GGACTACHVGGGTWTCTAAT-3′) [22,23]. Each PCR (25 µL) contained 12.5 µL of AccuStart II PCR ToughMix (final concentration 1×; QuantaBio, Beverly, MA, USA), 1 µL of DNA/cDNA template, 1 µL of each primer (final concentration 200 pM), and 9.5 µL of MOBIO PCR water (MOBIO, Carlsbad, CA, USA). PCR amplification was run for 35 cycles for DNA samples, and 23 cycles for RNA samples to minimize the accumulation of random errors during reverse transcription. Apart from this, all library preparation procedures were kept identical for DNA and cDNA samples. The PCR started with 3 min at 94 °C followed by 35 or 23 cycles at 94 °C for 45 s, 50 °C for 60 s, and 72 °C for 90 s, with a final extension at 72 °C for 10 min. Amplicons were pooled in equimolar concentrations and used for paired-end sequencing (2 × 151 bp) on an Illumina MiSeq platform (Illumina, Hayward, CA, USA) using the MiSeq reagent kit V2 [22]. Sequencing was performed in separate runs for DNA and cDNA samples at the Environmental Sample Preparation and Sequencing Facility of the Argonne National Laboratory, USA. Raw reads of both DNA- and RNA-based sequences used in this study are available in the Sequence Read Archive of the National Center for Biotechnology information under the accession number PRJNA546612.

### 2.3. Sequence Processing, Analysis of Community Structure and Statistical Analyses

We used the QIIME2 pipeline (version 2019.10) to process 16S rRNA gene and rRNA sequences [24]. The demultiplex sequences were uniformly trimmed to 150 bp (forward and reverse), and then were denoised to infer Amplicon Sequence Variants (ASVs) that were of 253 bp in length using the DADA2 plugin with default settings [25]. Because the error models might be different between runs, we performed a DADA2 denoising process on each run separately, and then merged feature tables and representative sequences from the two runs. Taxonomy was assigned to representative sequences using the Silva 132 Naive Bayes 515F/806R taxonomy classifier [26]. All ASVs affiliated to archaea, chloroplast and mitochondria, as well as singletons were removed from the dataset. A *de novo* phylogenetic tree was generated from representative sequences by aligning sequence fragments via MAFFT, masking ambiguous alignments and inferring a tree using the FastTree algorithm [27]. A rooted tree was created by putting root at the midpoint of the farthest tips among the tree. To make samples comparable, the feature table was rarefied to a depth of 15,000 sequences per sample. We estimated β-diversity using both the weighted and unweighted UniFrac distance in QIIME 2.

All subsequent analyses were carried out in R (v3.5.0) [28,29]. To compare bacterial communities between DNA- and RNA-based approaches, among successional stages and across time points, principal coordinate analysis (PCoA) and permutational multivariate analysis of variance (PERMANOVA) were conducted using the ‘pcoa’ and ‘adonis’ function in the packages ape and vegan, respectively [30,31]. Pair sample Wilcoxon tests were performed to compare the temporal turnover between the DNA- and RNA-inferred bacterial communities within each successional stage [30].

### 2.4. Quantification of Community Assembly Processes Governing Community Succession

We applied a framework that assesses the phylogenetic and taxonomic turnover of communities, and further used null modelling distributions to quantify the relative influences of distinct assembly processes mediating community turnover [19,32]. In the first step, we estimated the importance of stochasticity and selection using the β-nearest taxon index (βNTI) between pairs of communities. This index compares the observed phylogenetic turnover in species between a pair of communities and a null distribution. The observed phylogenetic turnover between pairs of communities was determined by β-mean nearest taxon distance (βMNTD) using function ‘comdistnt’ in the package picante [33]. The null distribution of phylogenetic turnover was generated by randomly shuffling the ASVs at the tip of the phylogenetic tree 999 times. As previously described [16,19], βNTI > 2 indicates variable selection is the dominant assembly process governing the turnover between a given pair of communities, since the phylogenetic turnover is significantly greater than that expected by chance. βNTI < -2 indicates homogeneous selection takes a leading role between a given pair of communities, as phylogenetic turnover is significantly lower than expected by chance. |βNTI| < 2 indicates the absence of selection, and a greater influence of stochastic processes, such as dispersal and/or drift. In the second step of the analysis, we examined the non-selection processes with the abundance weighted Raup-Crick metric (RC_bray_). We did this by comparing the ASV taxonomic turnover between a pair of communities and the null distribution [32,34]. To create a RC_bray_ metric, the Bray–Curtis dissimilarity between observed communities was first calculated. Then, the null distribution of the Bray–Curtis dissimilarity between simulated communities were constructed by randomly sampling ASVs 999 times. The RC_bray_ matrix was generated by comparing the Bray–Curtis dissimilarity between a pair of communities and the null distribution of Bray–Curtis dissimilarity. When |βNTI| < 2 and RC_bray_ > 0.95, community turnover is dominated by dispersal limitation, as the dissimilarity between observed communities is higher than the expectation. |βNTI| < 2 and RC_bray_ < -0.95 indicates homogenizing dispersal, as the community turnover between observed communities is lower than the null expectation. If |βNTI| < 2 and |RC_bray_| < 0.95, both phylogenetic and taxonomic community turnover of observed communities are not different from the null distributions. In other words, neither selection nor dispersal dominate the assembly processes, their influences on community turnover act together with drift, thus being termed as ‘undominated processes’. Together, we quantify the relative influence of each assembly process by the proportion of each assembly process within each dataset or treatment.

The analysis of βNTI relies on the correlation between relatedness of species phylogeny and their ecological niches. We estimated the optimal niche of ASVs (occurrence > 5) with soil pH, soil sodium concentration, soil organic carbon and soil water content using the function ‘wascores’ in the package vegan. Phylogenetic distance across ASVs was generated using the ‘cophenetic’ function in the package picante. Last, we tested the phylogenetic signal (i.e., the correlation between phylogenetic distances and the distances between optimal soil conditions across ASVs) using the function ‘mantel.correlog’ in the package vegan. For all the four soil parameters, significant positive correlations were observed at short phylogenetic distances, confirming the assumption for suitable ecological inferences using short phylogenetic distance based on βNTI (Appendix A).

Figures were made using the ggplot2 package [35]. All scripts used in this study are available on GitHub: https://github.com/Jia-Xiu/Jia_et_al_Microorganisms_2020.

## 3. Results 

### 3.1. Comparing Bacterial Community Structure Based on DNA and RNA Approaches

After denoising, filtering the low quality, short length and chimeric sequences, as well as removing singleton and non-target taxa, the total dataset consisted of 12,741,738 reads (this encompasses 120 bacterial 16S rRNA libraries, 60 for DNA- and 60 for RNA-inferred bacterial communities). After rarefying the feature table to 15,000 reads per sample, the rarefaction curves for most DNA-based samples reached a steady plateau, but some RNA-based samples did not, which indicates the sampling efforts were enough for most DNA-based samples, while more sequencing depth was potentially needed for some RNA-based samples (Appendix A). In the end, a total of 28,278 unique ASVs were obtained for the complete dataset, in which 19,608 ASVs were observed from the DNA-based dataset and 20,784 ASVs from the RNA-based dataset (Appendix A). We found the variation of α-diversity (i.e., richness, phylogenetic diversity, Shannon and Pielou’s evenness indexes) to be greater in RNA-inferred communities than in DNA-inferred communities across all successional stages (see Appendix A for details).

Principal coordinates analysis based on both weighted and unweighted UniFrac metrics showed a clear separation of bacterial communities in each successional stage at the first axis of PCoA, indicating the majority of the variation in β-diversity was attributed to differences across successional stages (Figure 2a,b). Sequencing approach (DNA- and RNA-based samples) had a lower effect on community composition than the successional stage. Even though community profiles based on DNA and RNA taken at the same successional stages were similar, a cluster separation of the DNA- and RNA-inferred bacterial communities was observed at the third axis of PCoA plots (Figure 2c–f). Accordingly, PERMANOVA results showed that successional stage was the most significant factor influencing the turnover of bacterial communities measured by both weighted and unweighted UniFrac metrics (*R*^2^ = 0.48 and *R*^2^ = 0.41, respectively; *p* < 0.001), followed by sequencing approach (*R*^2^ = 0.12 and *R*^2^ = 0.07, respectively; *p* < 0.001; Table 1 and Appendix A). We also observed the sampling month significantly contributes to the variation of bacterial communities, albeit at a smaller magnitude (*R*^2^ = 0.03, *p* < 0.001; Table 1 and Appendix A).

By further looking into the temporal variation of bacterial communities within each successional stage, we found the temporal turnover of RNA-inferred communities to be significantly higher than that of DNA-inferred communities across all successional stages (*p* < 0.001 in Wilcoxon signed-rank test, Figure 3). This pattern was further statistically supported by PERMANOVA (the influence of sampling time points on RNA-inferred communities: *R*^2^ = 0.051, *p* < 0.001; DNA-inferred communities: *R*^2^ = 0.038, *p* < 0.05; Appendix A). Across successional stages, we found the temporal turnover of bacterial communities to be higher at early successional stages and to progressively decrease as succession spans for both DNA- and RNA-based approaches (Figure 3). The influence of sampling month on community turnover was stronger at early (i.e., 0 and 10 years) than late successional stages (i.e., 40, 70 and 110 years; Appendix A).

We identified the dominant phyla (>3% of total abundance) in both DNA and RNA datasets to be Proteobacteria, Bacteroidetes, Actinobacteria, Acidobacteria and Planctomycetes (Figure 4). As ecological succession proceeds, the relative abundance of most bacterial phyla change in a similar manner in both DNA- and RNA-based samples. For instance, the relative abundance of Firmicutes were nearly identical in both datasets across all successional stages. However, even though most of the dominate phyla changed their relative abundances in a similar manner, their relative abundances in DNA- and RNA-based samples were different. Some phyla had higher relative abundances in the DNA dataset, such as Acidobacteria, Actinobacteria, Planctomycetes, Gemmatimonadetes and Verrucomicrobia, while others such as Proteobacteria and Entotheonellaeota had higher relative abundances in the RNA dataset. Besides, we also found different phyla distribution patterns across successional stages between DNA- and RNA-based samples. For example, Cyanobacteria only appeared at a higher relative abundance in the RNA-inferred communities at early successional stages, but not in the DNA-inferred communities. The relative abundance of Nitrospirae in the RNA-inferred communities decreased quickly along succession, but not in the DNA-inferred communities. The relative abundance of the candidate phylum V18 was stable in the RNA-inferred communities but declined rapidly at the DNA-inferred communities as succession proceeded.

### 3.2. Differences in Aseembly Processes between DNA- and RNA-Inferred Communities

We examined whether and how the interplay of assembly processes varies between DNA- and RNA-based samples by calculating the βNTI and the RC_bray_ metric. The results show that selection (i.e., variable and homogeneous) dominated the assembly processes of both DNA- and RNA-inferred communities (Figure 5a). Homogeneous selection was the dominant process across all samples, especially for the temporal variation, as succession proceeded (Figure 5b). Variable selection was overall less frequent, accounting for the temporal variation of communities at the initial successional stage (0 year of succession; Figure 5b and Appendix A) and the spatial turnover of communities at different sampling time points (Appendix A). Interestingly, in relative terms, we found homogeneous selection to have a stronger signal in the community assembly of DNA-inferred communities, whereas variable selection had a stronger signal in the RNA-inferred communities (Figure 5a). Similar results were also observed in the interplay of assembly processes accounting for both temporal and spatial variations of bacterial communities (Figure 5b and Appendix A).

## 4. Discussion 

In this study we examined the assembly processes underlying bacterial community succession by considering discrepancies in characterizing bacterial communities using DNA- and RNA-based approaches. Similar to previous studies [21], we found bacterial communities gradually change over time along this primary successional chronosequence based on both DNA and RNA community inferences. Patterns of temporal turnover of DNA- and RNA-inferred communities were also found to significantly change over the course of succession. We detected significantly higher temporal turnover at the early stages of succession, corroborating with previous findings in the ecological successional ecosystems [21,36]. Although community composition and dynamics were found to be similar between DNA- and RNA-based approaches, differences exist between both methods which is consistent what has been found in previous work [37]. Most importantly, the DNA- and RNA-inferred communities often displayed different dynamic changes, with RNA-inferred communities changing faster over time [38,39]. These differences are likely attributed to the distinct noises imposed by extracellular DNA and RNA due to differences in their stabilities and lifetime in the environment. In this salt-marsh ecosystem, the tides constantly bring microbial cells (i.e., dispersal) from sea and replenish the soil bacterial communities at initial stages. It is likely that maladapted organisms can rapidly die and significantly enrich the pool of environmental relic DNA, thus affecting the diversity estimation [11].

Moreover, observed differences between RNA- and DNA-inferred communities can occur due to intrinsic differences in the copy number and transcription of the 16S rRNA genes across distinct taxa. For example, we found Proteobacteria and Cyanobacteria to be detected at higher relative abundances in the RNA- rather than DNA-inferred communities, which corroborates the finding reported by Denef et al. [13]. Proteobacteria taxa are likely to have higher copies of the 16S RNA gene in their cells, which have been previously considered as copiotrophs [40,41]. However, in some cases, the copy number of ribosomes has nothing to do with cell activity. For instance, a Cyanobacteria species, *Aphanizomenon ovalisporum*, has a high number of ribosomes in its dormancy rather than in its vegetative cell [42]. Cell size was also suggested to positively correlate with the number of ribosomes in bacterial cells [13]. These discussions are not only relevant for bacterial communities, as fungal communities in groundwater aquifers were also found to have discrepant profiles when based on DNA and RNA approaches. For instance, 30–40% of the total fungal operational taxonomic units (OTUs) were only detected in RNA-based sequencing [43]. Taken together, the copy number of 16S rRNA gene, the metabolic state of a cell and innate ribosome content all can affect the disproportionate recovery of different bacteria based on DNA- and RNA-inferred community profiling.

DNA-based amplicon sequencing was previously found to inflate richness estimation, given that environmental DNA does not only encompasses viable bacteria cells but also relic DNA. However, our results show an opposite pattern. This discrepancy might occur because low-abundant bacteria that are metabolically active have higher chances of being detected using RNA- rather than DNA-based gene sequencing, since metabolic active (high growth rate) bacterial cells contain more ribosomes than inactive cells [44]. In this study, we observed more unique rare taxa in the RNA dataset compared to the DNA dataset (Appendix A). In addition, a substantial number of rare taxa were detected in higher rRNA:rDNA ratios (Appendix A), which suggest they were metabolically active. In line with our expectation, a study in glacier-fed streams reported that low abundant taxa were over-present in the community profiled by 16S rRNA (cDNA) sequencing [45]. In this context, the higher richness of taxa in the RNA dataset is reasonable and to some extent expected, since active rare taxa that are not identified by DNA-based sequencing can be detected using the RNA-based approach. Given that metabolically active rare taxa can disproportionally be detected when using these two approaches, we advise that future studies focusing on rare taxa should take a careful consideration of data interpretation when based on DNA and RNA sequencing inferences.

The quantification of community assembly processes using both DNA- and RNA-based approaches was dominated by selection. In relative terms, the influence of variable selection was higher in the RNA-inferred communities, suggesting that RNA-inferred communities more rapidly respond to environmental fluctuations compared to a more ‘stable’ scenario of DNA-based communities. This is consistent with a previous finding showing that the phylogenetic community structure of RNA-based communities changes quickly in response to variations in pH and carbon to nitrogen ratios [18]. Moreover, the higher influence of homogenous selection in the DNA-based communities indicates that communities profiled by DNA are expected to display higher overall correlations with stringent environmental factors that are homogeneously distributed. In addition, the direct use of DNA (particularly in systems such as soils) can often account for a large proportion of inactive cells/taxa that has no environmental responses. This aligns with the idea that cell inactivation and dormancy constitute a life strategy to persist in the environment under unfavorable conditions [46]. Together, since DNA- and RNA-based sequencing have different outcomes in recovering bacterial communities, inferences in the quantitative influence of community assembly processes inferred by null modelling analysis will also likely vary significantly.

## 5. Conclusions

The broad use of amplicon sequencing has greatly advanced our understanding of the ecological processes structuring bacterial communities during succession. However, DNA- and RNA-based approaches can generate distinct profiles of community composition. This can be caused by differences in the stability of DNA and RNA, differences in the copy numbers of the 16S rRNA gene, in addition to changes in the number of transcripts of this gene, i.e., differences in cell active states and lifestyle. Here, we used these two approaches to profile bacterial communities across a primary successional gradient and quantified the relative influences of community assembly processes governing community turnover. Our results demonstrate that RNA-based communities have greater variation in community composition and are relatively more influenced by variable selection; while DNA-inferred communities have less variation and are relatively more influenced by homogeneous selection. Future studies advancing knowledge on the community assembly of bacterial communities and successional dynamics must be cautious when interpreting data obtained from either DNA- or RNA-based sequencing approaches.

## Figures and Tables

**Figure 1 microorganisms-08-00798-f001:**
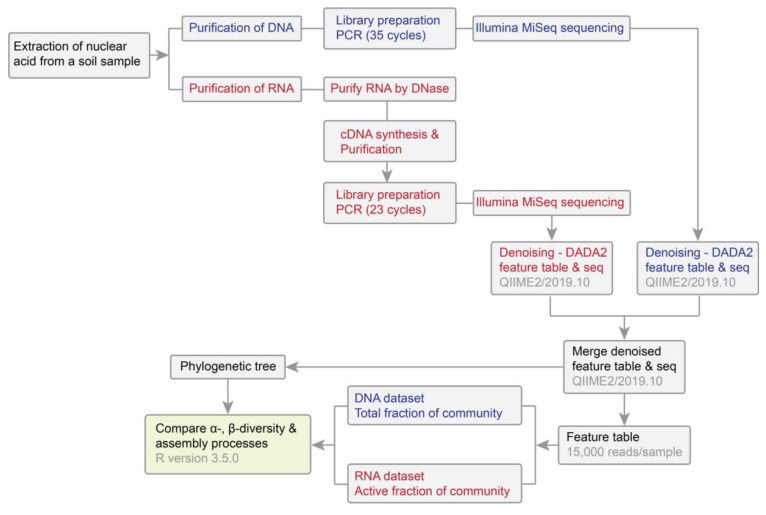
Workflow applied in this study. Procedures for analyzing RNA samples are shown in red, procedures for analyzing DNA samples are shown in blue, and common procedures are shown in black.

**Figure 2 microorganisms-08-00798-f002:**
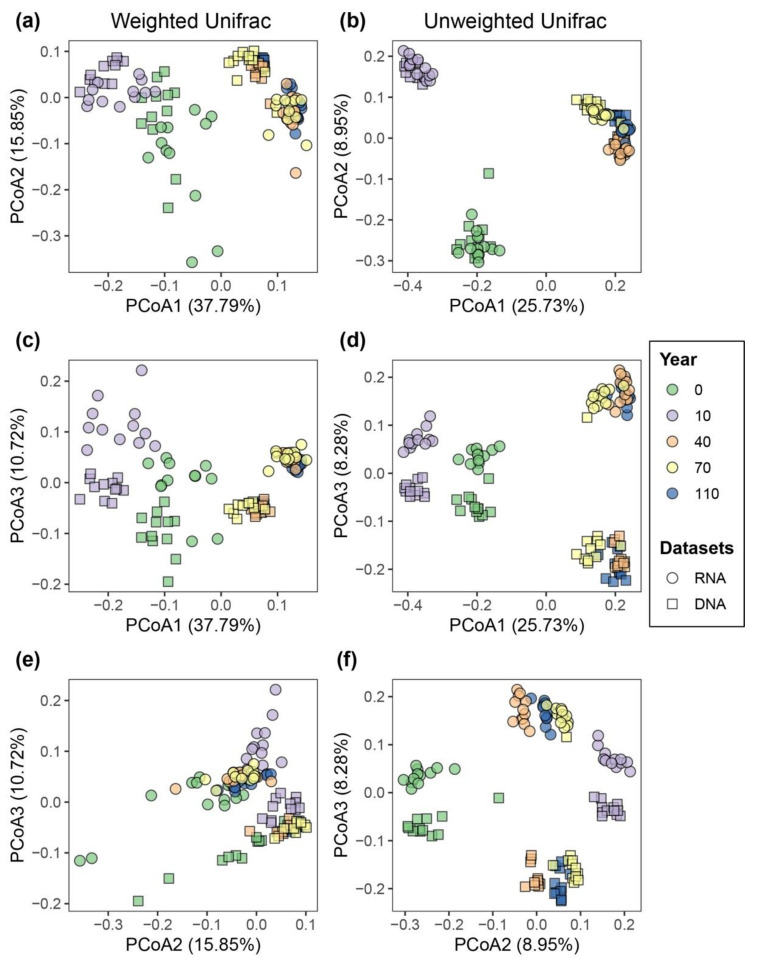
β-diversity patterns of bacterial communities displaying differences according to soil successional stages and sequencing approach (i.e., DNA- or RNA-based). Weighted (**a**,**c**,**e**) and unweighted (**b**,**d**,**f**) UniFrac distances were used to calculate β-diversity, which was visualized using principal coordinate analysis (PCoA). PCoA results are illustrated in different ordinate axes. Colors represent successional stages (i.e., 0, 10, 40, 70 and 110 years of succession), and different shapes represent RNA- and DNA-based approaches. The percentages in the axes show the variation of species composition explained by each ordinate.

**Figure 3 microorganisms-08-00798-f003:**
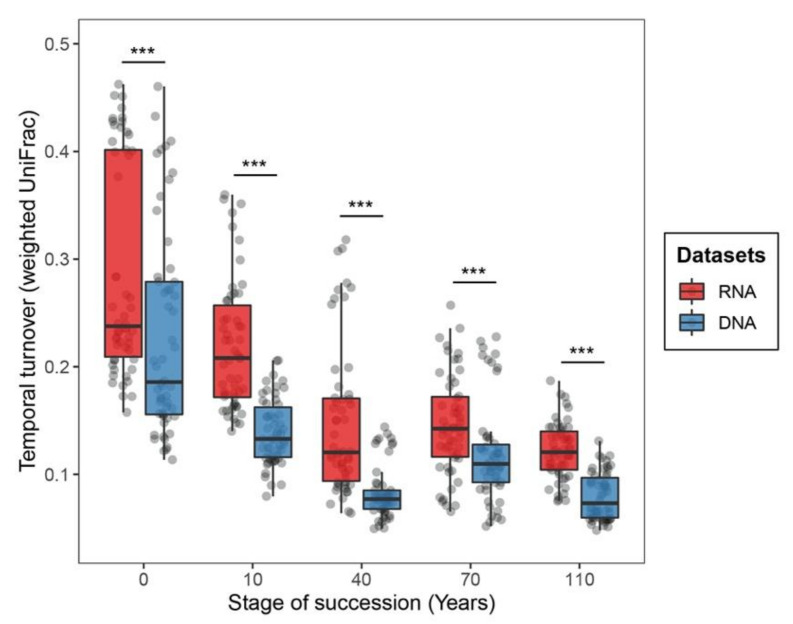
Boxplots displaying the temporal variation of RNA- and DNA-inferred communities along the soil successional stages. The temporal variation was based on weighted UniFrac distances of communities across distinct sampling time points. *** indicates *p* < 0.001 in Wilcoxon signed-rank test.

**Figure 4 microorganisms-08-00798-f004:**
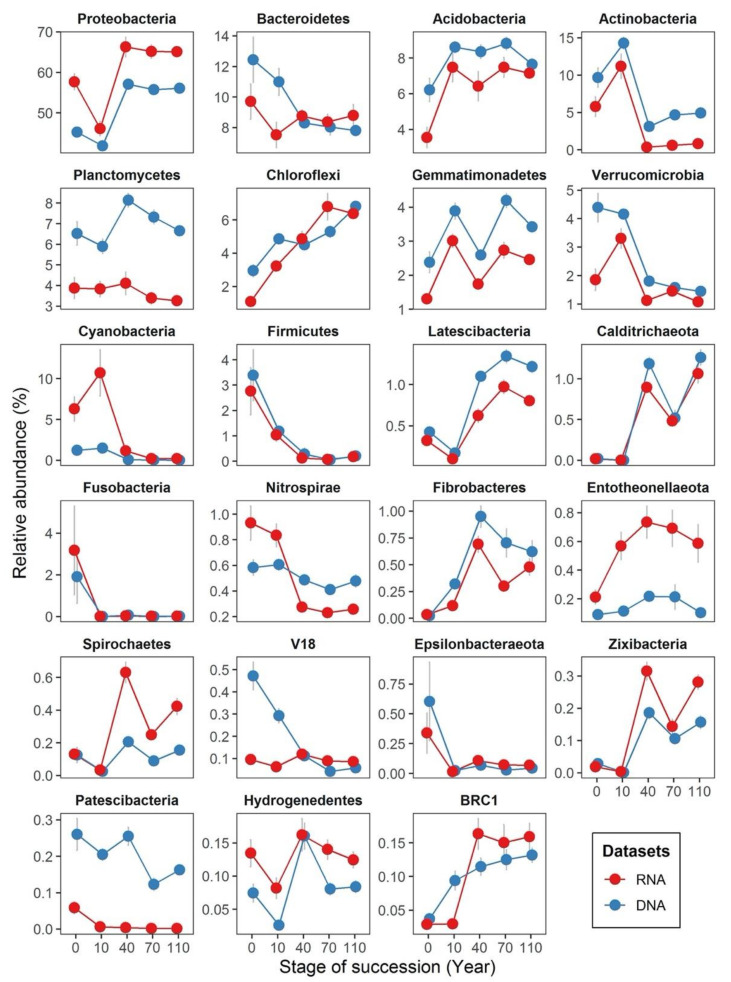
Line plots displaying the dynamic changes in relative abundances of bacterial phyla (mean relative abundance > 0.1%) based on DNA- and RNA-inferred communities. The *x*-axis displays successional stages in years (i.e., 0, 10, 40, 70 and 110 years of succession), and the *y*-axis displays the phyla relative abundance (in % of the total abundance). Points indicate average, and error bars represent standard errors from the averages.

**Figure 5 microorganisms-08-00798-f005:**
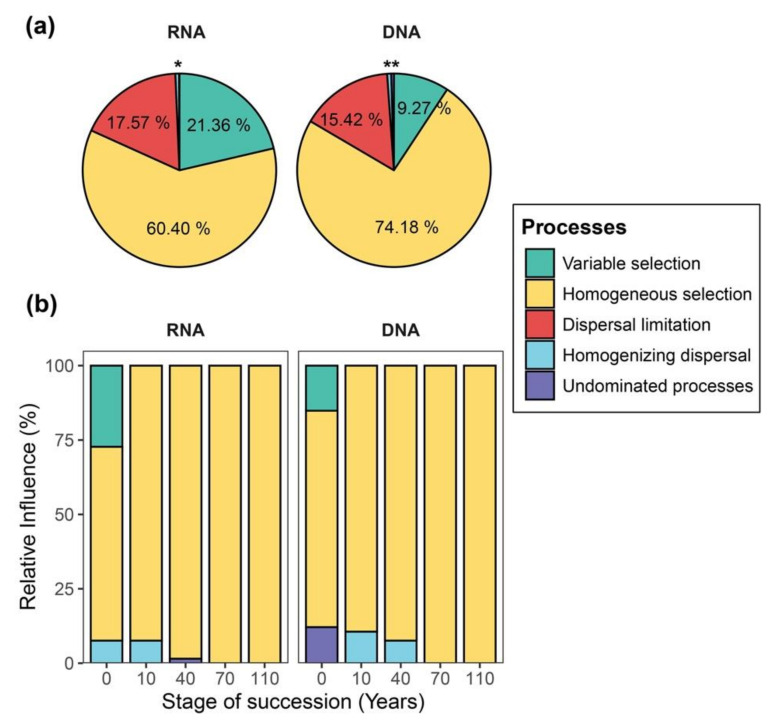
Pie charts and stacked-bar plots displaying the relative influences of assembly processes governing community turnover. (**a**)The pie charts show the relative influence of distinct assembly processes determining the spatiotemporal variation of RNA- and DNA-inferred communities. (**b**) Stacked-bar plots display the relative influences of distinct assembly processes structuring the temporal variation of bacterial communities in each successional stage based on both RNA- and DNA-based approaches. * indicates the influence of homogenizing dispersal and undominated processes for the turnover of RNA-inferred communities to be 0.62% and 0.06%, respectively. ** indicates the influence of homogenizing dispersal and undominated processes for the turnover of DNA-inferred community to be 0.68% and 0.45%, respectively.

**Table 1 microorganisms-08-00798-t001:** Three-way permutational multivariate analysis of variance (PERMANOVA) showing the influence of different factors on β-diversity of bacterial communities based on weighted UniFrac distances. The rows ‘Dataset’, ‘Year’ and ‘Month’ indicate RNA- or DNA-based approaches, successional stages and within-stage temporal turnover, respectively.

Groups	Df *	SumSqs *	MeanSqs *	F.Model *	*R*^2^ *	*p*-Values ^†^
**Dataset**	1	0.605792	0.605792	65.36243	0.123029	**<0.001**
**Year**	4	2.373268	0.593317	64.01644	0.481983	**<0.001**
**Month**	3	0.164918	0.054973	5.931321	0.033493	**<0.001**
**Dataset:Year**	4	0.309029	0.077257	8.335727	0.06276	**<0.001**
**Dataset:Month**	3	0.047992	0.015997	1.72604	0.009747	0.0436
**Year:Month**	12	0.523401	0.043617	4.706062	0.106297	**<0.001**
**Dataset:Year:Month**	12	0.158108	0.013176	1.421602	0.03211	0.0246
**Residuals**	80	0.741456	0.009268		0.150581	
**Total**	119	4.923963			1	

* Df—degrees of freedom; SumSq—sum of squares; MeanSqs—mean of squares; F.Model—F value by permutation; *R*^2^—explained variation; *p*-values based on 9999 permutations. † significant *p*-values (*p* < 0.001) are shown in bold.

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
