# Peer review of "Comparing the Influence of Assembly Processes Governing Bacterial Community Succession Based on DNA and RNA Data"

_microorganisms, 2020, doi:10.3390/microorganisms8060798_

Round 1

Reviewer 1 Report

Within the manuscript from Jia et al., the dynamics of bacterial community composition across five successional stages in a primary soil chronosequence were analyzed based on 16S rRNA genes and transcripts. Findings revealed differences between 16S rRNA gene and transcript-based analyses, an aspect which should also be considered when designing and interpreting future studies. In general, the manuscript is logically organized and collected data are analyzed in a coherent way and support the conclusions drawn. Nevertheless, the following comments need to be addressed by the authors:  

(Pg 2 Ln 53-55: “However, few studies have examined bacterial community succession using both DNA- and RNA-based approaches [9]” - only one reference is considered here - if possible, the authors should include one or two further reference(s)

Pg 3 Ln 112-113: “Remaining DNA in RNA samples was removed using the DNase Max kit (Qiagen, Germany).” - did the authors verify the successful removal of DNA (for instance by performing PCR reactions targeting the 16S rRNA gene)?   

The authors wrote (Pg 3 Ln 134-135): “We used QIIME2 (version 2019.10) to demultiplex sequences, and then the DADA2 plugin to denoise and generate Amplicon Sequence Variants (ASVs) that were of 253 bp in length [17]” - here, the authors only provided the reference for QIIME2, but they should also provide the reference for DADA2 (Callahan, B.J.; McMurdie, P.J.; Rosen, M.J.; Han, A.W.; Johnson, A.J.; Holmes, S.P. Dada2: high-resolution sample inference from Illumina amplicon data. Nature methods 2016, 13, 581-583.) - furthermore, the authors should specify if they used this QIIME2 plugin with default or modified settings

There is only one table in the main manuscript - this table has been designated “Table 2” - please change “Table 2” to “Table 1”

Further comments:

Abstract

- Pg 1 Ln 18: please change “…that inferred by DNA…” to “…those inferred by DNA …”

- Pg 1 Ln 21: please change “…16S rRNA gene…” to “…16S rRNA genes…”

- Pg 1 Ln 23: please change “…quantify…” to “…quantified…”

- Pg 1 Ln 28: please change “…which largely influenced…” to “…which was largely influenced…”

Introduction

- Pg 1 Ln 35: please change “…and their effects on ecosystem functioning…” to “…and affect ecosystem functioning …”

- Pg 1 Ln 40: please change “…16S rRNA gene…” to “…16S rRNA genes…”

- Pg 2 Ln 59: please change “…has been…” to “…have been…”

- Pg 2 Ln 62: please change “…dynamic…” to “…dynamics…”

- Pg 2 Ln 65: please change “…has shown that a strong…” to “…has shown a strong…”

- Pg 2 Ln 73-74: please change “…resulting differences on community turnover and variation…” to “…resulting in community turnover and variation differences…”

- Pg 2 Ln 76: please change “…that inferred by DNA.” to “…those inferred by DNA.”

- Pg 2 Ln 79: please change “…dynamics on bacterial community composition…” to “…dynamics of bacterial community composition…”

Materials and Methods

- Pg 3 Ln 94: please change “This natural chronosequences spans…” to “This natural chronosequence spans…”

- Pg 3 Ln 124: please change “…to minimized…” to “…to minimize…”

- Pg 4 Ln 153: please change “…processses…” to “…to processes …”

- Pg 4 Ln 154: please change “…assess…” to “…assesses…”

- Pg 5 Ln 193: please change “All scripts used in this study is available…” to “All scripts used in this study are available …”

Results

- Pg 5 Ln 203: please change “After rarefying feature table…” to “After rarefying the feature table …”

- Pg 5 Ln 218: please change “…PCoA plot…” to “…PCoA plots…”

- Pg 6 Ln 225: please change “…bacterial community…” to “…bacterial communities…”

- Pg 6 Ln 231: please change “…decreased…” to “…decrease…”

- Pg 8 Ln 267: please change “…candidate phyla V18…” to “…candidate phylum V18…”

Discussion

- Pg 11 Ln 318-319: please change “Proteobacteria taxa is likely to have higher copies of the 16S RNA gene in their cells, which has been previously considered as copiotrophs…” to “Proteobacteria taxa are likely to have higher copies of the 16S RNA gene in their cells, which have been previously considered as copiotrophs…”

- Pg 11 Ln 319: please change “…However, in some case…” to “…However, in some cases …”

- Pg 11 Ln 321 and also Ln 322: please change “…number of ribosome…” to “…number of ribosomes…”

- Pg 11 Ln 325: please change “For instance, such as 30-40%…” to “…For instance, 30-40%…”

- Pg 11 Ln 359: please change “…will also varies significantly.” to “…will also vary significantly.”

Conclusions

- Pg 11/12 Ln 364-365: please change “…the ribosome 16S rRNA gene…” to “…the 16S rRNA gene …”

Figures

- Figure 1: in one of the boxes “Extractraction of nuclear acid from a soil sample” is written - please change this to “Extraction of nucleic acids from a soil sample”; with respect to the legend, please change “…are showed in red, while procedures for analyzing DNA samples are showed in blue…” to “…are shown in red, while procedures for analyzing DNA samples are shown in blue…”  

- Figure 2: with respect to the legend please change “Unifrac distances was used to calculate…” to “Unifrac distances were used to calculate…” and please change “The percentages in the axes shows…” to “The percentages depicted at the axes show…”  

- Figure 5: with respect to the explanation of the colors it seems that “Dispersal limilation” instead of “Dispersal limitation” has been written - please change this

Author Response

Response to Reviewer 1 Comments

General comments: Within the manuscript from Jia et al., the dynamics of bacterial community composition across five successional stages in a primary soil chronosequence were analyzed based on 16S rRNA genes and transcripts. Findings revealed differences between 16S rRNA gene and transcript-based analyses, an aspect which should also be considered when designing and interpreting future studies. In general, the manuscript is logically organized and collected data are analyzed in a coherent way and support the conclusions drawn. Nevertheless, the following comments need to be addressed by the authors:

Response: We thank the reviewer the positive feedback provided.

Point 1: Pg 2 Ln 53-55: “However, few studies have examined bacterial community succession using both DNA- and RNA-based approaches [9]” - only one reference is considered here - if possible, the authors should include one or two further reference(s)

Response 1: Two additional references were added, i.e. Jurburg et al. 2017 and Denef et al. 2016. Please see Ln 55.

Point 2: Pg 3 Ln 112-113: “Remaining DNA in RNA samples was removed using the DNase Max kit (Qiagen, Germany).” - did the authors verify the successful removal of DNA (for instance by performing PCR reactions targeting the 16S rRNA gene)?

Response 2: Thank you for pointing this out. We did perform PCR reactions to verify the successful removal of DNA from RNA samples. We have checked 10% of RNA samples to verify whether the amount of DNase applied was enough to ensure DNA to be completely removed from RNA samples. Additional clarification on this issue is now provided in the Material and Methods section. Please see Ln 113-115.

Point 3: The authors wrote (Pg 3 Ln 134-135): “We used QIIME2 (version 2019.10) to demultiplex sequences, and then the DADA2 plugin to denoise and generate Amplicon Sequence Variants (ASVs) that were of 253 bp in length [17]” - here, the authors only provided the reference for QIIME2, but they should also provide the reference for DADA2 (Callahan, B.J.; McMurdie, P.J.; Rosen, M.J.; Han, A.W.; Johnson, A.J.; Holmes, S.P. Dada2: high-resolution sample inference from Illumina amplicon data. Nature methods 2016, 13, 581-583.) - furthermore, the authors should specify if they used this QIIME2 plugin with default or modified settings

Response 3: The indicated reference was added (Ln 139). We used the DADA2 plugin with default settings. Please see the modified text in Ln 136-139.

Point 4: There is only one table in the main manuscript - this table has been designated “Table 2” - please change “Table 2” to “Table 1”

Response 4: Correction made.

Further comments:

Abstract

Point 5: Pg 1 Ln 18: please change “…that inferred by DNA…” to “…those inferred by DNA …”

Response 5: Correction made. Please see Ln 18.

Point 6: Pg 1 Ln 21: please change “…16S rRNA gene…” to “…16S rRNA genes…”

Response 6: Correction made. Please see Ln 22.

Point 7: Pg 1 Ln 23: please change “…quantify…” to “…quantified…”

Response 7: Correction made. Please see Ln 23.

Point 8: Pg 1 Ln 28: please change “…which largely influenced…” to “…which was largely influenced…”

Response 8: Correction made. Please see Ln 28.

Introduction

Point 9: Pg 1 Ln 35: please change “…and their effects on ecosystem functioning…” to “…and affect ecosystem functioning …”

Response 9: Correction made. Please see Ln 35.

Point 10: Pg 1 Ln 40: please change “…16S rRNA gene…” to “…16S rRNA genes…”

Response 10: Correction made. Please see Ln 40.

Point 11: Pg 2 Ln 59: please change “…has been…” to “…have been…”

Response 11: Correction made. Please see Ln 59.

Point 12: Pg 2 Ln 62: please change “…dynamic…” to “…dynamics…”

Response 12: Correction made. Please see Ln 62.

Point 13: Pg 2 Ln 65: please change “…has shown that a strong…” to “…has shown a strong…”

Response 13: Correction made. Please see Ln 65-66.

Point 14: Pg 2 Ln 73-74: please change “…resulting differences on community turnover and variation…” to “…resulting in community turnover and variation differences…”

Response 14: Correction made. Please see Ln 73-74.

Point 15: Pg 2 Ln 76: please change “…that inferred by DNA.” to “…those inferred by DNA.”

Response 15: Correction made. Please see Ln 76.

Point 16: Pg 2 Ln 79: please change “…dynamics on bacterial community composition…” to “…dynamics of bacterial community composition…”

Response 16: Correction made. Please see Ln 79.

Materials and Methods

Point 17: Pg 3 Ln 94: please change “This natural chronosequences spans…” to “This natural chronosequence spans…”

Response 17: Correction made. Please see Ln 94.

Point 18: Pg 3 Ln 124: please change “…to minimized…” to “…to minimize…”

Response 18: Correction made. Please see Ln 126.

Point 19: Pg 4 Ln 153: please change “…processses…” to “…to processes …”

Response 19: Correction made. Please see Ln 157.

Point 20: Pg 4 Ln 154: please change “…assess…” to “…assesses…”

Response 20: Correction made. Please see Ln 158.

Point 21: Pg 5 Ln 193: please change “All scripts used in this study is available…” to “All scripts used in this study are available …”

Response 21: Correction made. Please see Ln 197.

Results

Point 22: Pg 5 Ln 203: please change “After rarefying feature table…” to “After rarefying the feature table …”

Response 22: Correction made. Please see Ln 208.

Point 23: Pg 5 Ln 218: please change “…PCoA plot…” to “…PCoA plots…”

Response 23: Correction made. Please see Ln 223.

Point 24: Pg 6 Ln 225: please change “…bacterial community…” to “…bacterial communities…”

Response 24: Correction made. Please see Ln 230.

Point 25: Pg 6 Ln 231: please change “…decreased…” to “…decrease…”

Response 25: Correction made. Please see Ln 237.

Point 26: Pg 8 Ln 267: please change “…candidate phyla V18…” to “…candidate phylum V18…”

Response 26: Correction made. Please see Ln 273.

Discussion

Point 27: Pg 11 Ln 318-319: please change “Proteobacteria taxa is likely to have higher copies of the 16S RNA gene in their cells, which has been previously considered as copiotrophs…” to “Proteobacteria taxa are likely to have higher copies of the 16S RNA gene in their cells, which have been previously considered as copiotrophs…”

Response 27: Correction made. Please see Ln 326-327.

Point 28: Pg 11 Ln 319: please change “…However, in some case…” to “…However, in some cases …”

Response 28: Correction made. Please see Ln 328.

Point 29: Pg 11 Ln 321 and also Ln 322: please change “…number of ribosome…” to “…number of ribosomes…”

Response 29: Correction made. Please see Ln 329.

Point 30: Pg 11 Ln 325: please change “For instance, such as 30-40%…” to “…For instance, 30-40%…”

Response 30: Correction made. Please see Ln 333.

Point 31: Pg 11 Ln 359: please change “…will also varies significantly.” to “…will also vary significantly.”

Response 31: Correction made. Please see Ln 367.

Conclusions

Point 32: Pg 11/12 Ln 364-365: please change “…the ribosome 16S rRNA gene…” to “…the 16S rRNA gene …”

Response 32: Correction made. Please see Ln 372.

Figures

Point 33: Figure 1: in one of the boxes “Extractraction of nuclear acid from a soil sample” is written - please change this to “Extraction of nucleic acids from a soil sample”; with respect to the legend, please change “…are showed in red, while procedures for analyzing DNA samples are showed in blue…” to “…are shown in red, while procedures for analyzing DNA samples are shown in blue…”

Response 33: Correction made accordingly. Please, see Figure 1 and the respective caption in Ln 199-202.

Point 34: Figure 2: with respect to the legend please change “Unifrac distances was used to calculate…” to “Unifrac distances were used to calculate…” and please change “The percentages in the axes shows…” to “The percentages depicted at the axes show…”

Response 34: These sentences were revised as suggested. Please see Ln 249 and 252.

Point 35: Figure 5: with respect to the explanation of the colors it seems that “Dispersal limilation” instead of “Dispersal limitation” has been written - please change this

Response 35: Correction made accordingly. Please see Figure 5.

Reviewer 2 Report

Dear authors, 

I liked the manuscript by Jia and colleagues that compares microbial diversity in soil samples taken over time. The analysis of diversity using both DNA and RNA based approaches is done and presented well. The analysis methods are understandable and the results are clear.

I have a couple of comments:

  1. How does this approach compare against metagenomics approaches that use:
    1. whole-genome shotgun sequencing; or
    2. long read DNA and RNA sequencing (PacBio or Oxford Nanopore).
  2. Would it be useful to use additional information from microbial 23S rRNA sequence for improving classification?
  3. Similar to 2., how about sequencing 18S for fungal species which would be important for analyzing diversity?

My best wishes to the authors. 

Cheers.

Author Response

Response to Reviewer 2  Comments

General comments: I liked the manuscript by Jia and colleagues that compares microbial diversity in soil samples taken over time. The analysis of diversity using both DNA and RNA based approaches is done and presented well. The analysis methods are understandable and the results are clear.

I have a couple of comments:

Response: We thank the reviewer for the positive feedback provided.

We want to clarify that some of the comments, questions and suggestions provided (see below) are not clearly in line with the scope of our manuscript, although they constitute interesting points for discussion for future studies. Please, see our detailed responses below.

Point 1: How does this approach compare against metagenomics approaches that use:

    1. whole-genome shotgun sequencing; or
    2. long read DNA and RNA sequencing (PacBio or Oxford Nanopore).

Response 1:

  • Metagenomic data are not adequate for the null modelling analyses as presented in this study. In particular, our model integrates phylogenetic information with permutation to investigate patterns of community deviation. For that end, amplicon data is required. Even though 16S rRNA sequences can be extracted from metagenomic data, this method can lead to an underestimation of diversity in bacterial communities, as low abundant taxa are not detected due to insufficient depth of shotgun sequencing approaches.
  • We expect that differences in the relative influences of distinct assembly processes when using DNA and RNA data will be similar for short- and long-read sequencing approaches. Differences might only emerge if read length causes an unlikely significant change in the overall structure of the phylogenetic tree. Also, another point of debate is the fact that current long-read sequencing technologies, such as PacBio or Oxford Nanopore, do not provide enough sequencing depth to detect rare taxa. As such, differences might also emerge as an artefact of drastic differences in sequencing depth across distinct datasets and data types.

Point 2: Would it be useful to use additional information from microbial 23S rRNA sequence for improving classification?

Response 2: Currently, it is a standard in the field to use the bacterial 16S rRNA gene to profile and analyse patterns of bacterial communities across divergent systems. The use of the 23S rRNA gene could potentially enhance classification, however, there are still no database available or community effort toward the development of this resource.

Point 3: Similar to 2., how about sequencing 18S for fungal species which would be important for analyzing diversity?

Response 3: We agree with the reviewer. If the paper was focused on patterns of community assembly of fungal communities, the most useful marker gene would be the 18S RNA gene. Given the nature of our model (which is based on phylogenetic information), ITS sequences would not provide robust phylogenetic information, and, as such, must be avoided. Worth mentioning, a previous study has found differences in fungal community composition that were profiled using RNA- and DNA- based sequencing (Ln 331-333 in our manuscript). We speculate that similar patterns are likely to be observed for fungal communities, i.e. stronger variable selection in RNA-inferred fungal communities, while stronger homogeneous selection in DNA-inferred fungal communities. However, caution is wanted in direct assuming these predictions as the overall structure of 18S RNA-based phylogeny is different from that of the 16S RNA gene. This occur because the rate of polymorphism in these genes are different, which direct influence the resolution in which species are defined.